# Engineering *Photorhabdus luminescens* toxin complex (PTC) into a recombinant injection nanomachine

Peter Njenga Ng'ang'a[1,2,3], Julia K Ebner[1,2,3], Matthias Plessner[1], Klaus Aktories[1,4], Gudula Schmidt[1]

**Engineering delivery systems for proteins and peptides into mammalian cells is an ongoing challenge for cell biological studies as well as for therapeutic approaches.** *Photorhabdus luminescens* **toxin complex (PTC) is a heterotrimeric protein complex able to deliver diverse protein toxins into mammalian cells. We engineered the syringe-like nanomachine for delivery of protein toxins from different species. In addition, we loaded the highly active copepod luciferase** *Metridia longa* **M-Luc7 for accurate quantification of injected molecules. We suggest that besides the probable size limitation, the charge of the cargo also influences the efficiency of packing and transport into mammalian cells. Our data show that the PTC constitutes a powerful system to inject recombinant proteins, peptides, and potentially, other molecules into mammalian cells. In addition, in contrast to other protein transporters based on pore formation, the closed, compact structure of the PTC may protect cargo from degradation.**

## Introduction

*Photorhabdus luminescens* is an entomopathogenic bacterium. It produces a large, heterotrimeric *Photorhabdus luminescens* toxin complex (PTC) assembled by three different components: TcA, TcB, and TcC which form a syringe-like injection apparatus preloaded with a toxin (Waterfield et al, 2005; Sheets & Aktories, 2017). Homologues of the PTC have been identified in other insect and human pathogenic bacteria (Hinchliffe et al, 2010). Different from the type-III secretion machineries of *Yersinia spp.* (see below), the *Photorhabdus* megadalton protein complex allows the delivery of its cargo in the absence of the bacteria. Injection of the cargo by the standalone toxin complex requires initial receptor binding on the target cell membrane. A shift to higher or to lower pH triggers structural rearrangements (Gatsogiannis et al, 2013; Meusch et al, 2014). The toxin complex is functional on insects, as well as on mammalian cells (Hares et al, 2008; Lang et al, 2010). The acidic pH within early endosomes is sufficient to allow toxin injection through the endosomal membrane into the cytosol. The molecular rearrangement of the Tc structure has been analysed in detail by cryoelectron microscopy: TcA is a pentamer forming an α helical channel surrounded by a shell-like structure (Gatsogiannis et al, 2013). Channel and shell are connected by a proline rich relaxed peptide, which contracts to a partially helical fold upon cell contact and acidification, thereby pushing the channel through the membrane and releasing the preloaded toxin into the cytosol (Gatsogiannis et al, 2016). TcB and TcC together form a cocoon-like structure (BC), which interacts with the TcA pentamer (Meusch et al, 2014; Roderer et al, 2019 *Preprint*). TcC encodes a toxic enzyme in its C-terminal hypervariable region (hvr) that is later encased by the BC cocoon (Gatsogiannis et al, 2016). Binding of BC to the TcA pentamer opens a gate, forming a joint channel for loading the toxic enzyme. TcC provides the hvr and an aspartyl autoprotease necessary to cleave off the enzyme (hvr) for its release into the channel (Meusch et al, 2014). The modular composition of the toxin complex allows loading and injection of various *Photorhabdus* toxins into mammalian cells presenting it as a possible vehicle for transport and injection of diverse enzymes and peptides of foreign origin. Here, we engineered the PTC (TcA + TcB – TcC) to allow delivery of *Yersinia enterocolitica* YopT, a typical type–III secreted toxin, into HeLa cells.

*Y. enterocolitica* is a food-borne pathogen causing acute and chronic gastroenteric infections in humans. The bacterium injects several toxins into mammalian cells by a syringe-like type-III secretion system upon direct contact of the bacteria with the eukaryotic cell (Boyd et al, 2000). The toxins, including the protease YopT (*Yersinia* outer protein T), are encoded on a virulence plasmid (pYV) (Iriarte & Cornelis, 1998). YopT is a cysteine protease that specifically cleaves the isoprenylated C terminus of Rho GTPases (Shao et al, 2002). Rho proteins are posttranslationally modified at their C-terminal CaaX-box by isoprenylation of the cysteine (C), truncation of -aaX (a-aliphatic, X-any amino acid) and methylation (Zhang & Casey, 1996). This modification is required for localization at cellular membranes and for transport by the guanosine nucleotide dissociation inhibitor (GDI) (Garcia-Mata et al, 2011). By

[1]Institute for Experimental and Clinical Pharmacology and Toxicology, Faculty of Medicine, University of Freiburg, Freiburg, Germany   [2]Faculty of Biology, University of Freiburg, Freiburg, Germany   [3]Spemann Graduate School for Biology and Medicine, University of Freiburg, Freiburg, Germany   [4]BIOSS Centre for Biological Signalling Studies, University of Freiburg, Freiburg, Germany

Correspondence: gudula.schmidt@pharmakol.uni-freiburg.de

removing the isoprenylated cysteine, YopT leads to the release of Rho proteins from cellular membranes and liberation from GDI (Aepfelbacher et al, 2003; Shao et al, 2003). Rho proteins are crucial regulators of important signalling pathways ranging from actin-dependent migration and immune cell function to survival (Nobes & Hall, 1994; Castellano et al, 2001). More specifically, YopT blocks phagocytic cup formation, chemotaxis, and proliferation (Iriarte & Cornelis, 1998; Aepfelbacher et al, 2003; Trulzsch et al, 2004). Functional injection of several YopT-based proteins allowed us to define the requirements for foreign proteins transported by the PTC. For accurate quantification, we additionally cloned the highly active secreted luciferase of the marine copepod *Metridia longa* as reporter protein into the TcB-TcC3 (BC3) complex because the enzyme meets all pre-conditions for delivery identified in our analysis.

# Results

## Generation of the recombinant toxin chimera BC3-C3bot

It was shown previously that in the toxin complex, the enzymatic component to be injected (hvr) is cleaved within the cocoon (Meusch et al, 2014). The two toxins of the natural *P. luminescens* effector cocktail so far analysed are ADP-ribosyltransferases (TcC3 and TcC5). As proof of concept, we asked whether we could exchange the hvr of TcC3 with C3bot (ADP-ribosyltransferase of *Clostridium botulinum*) for functional transport of the *Clostridium* ADP-ribosyltransferase into mammalian cells (Aktories & Koch, 1997). Therefore, C3bot was cloned in place of the original TcC3hvr, that is, C-terminal to the auto-proteolytic cleavage site of truncated BC3 (functional cocoon-forming BC fusion protein of *P. luminescens*

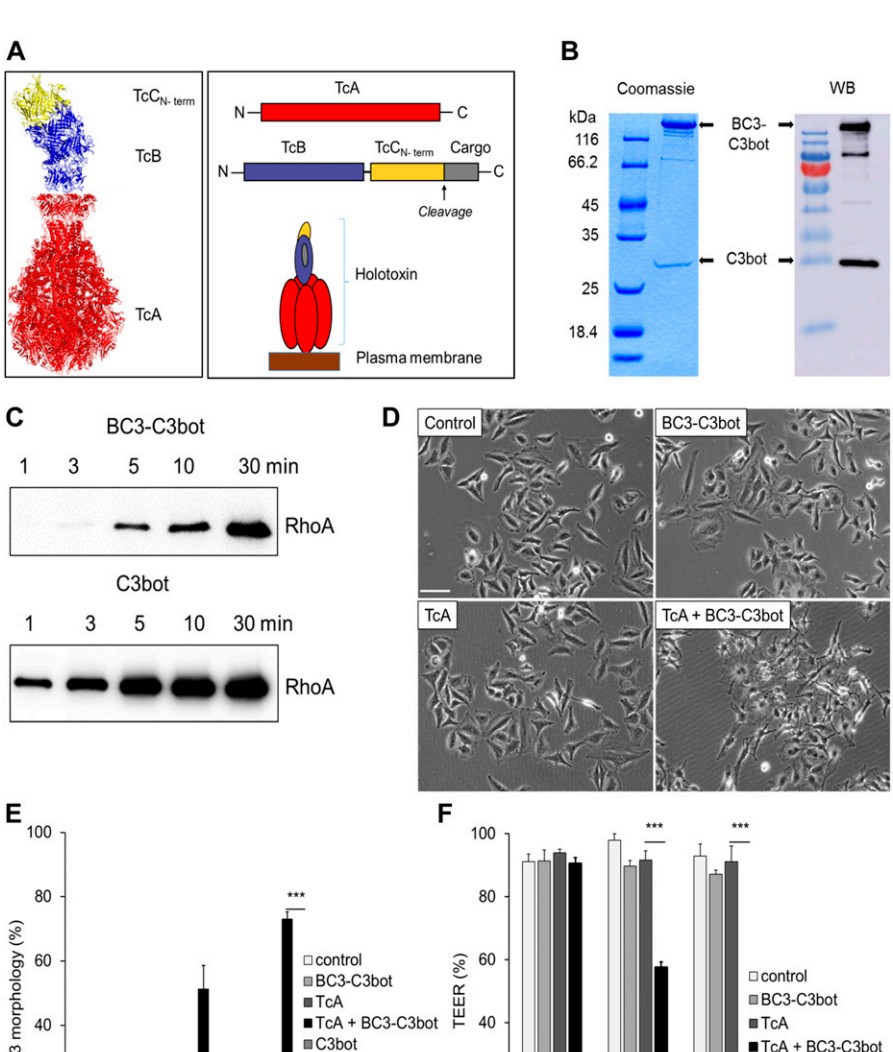

**Figure 1. Activity of BC3-C3bot fusion toxin.**
**(A)** Design of the fusion toxin. The TcC3 C-terminal hvr was replaced with a selected cargo C terminus to the auto proteolytic cleavage site (grey box). This cargo would then be packaged into the BC3 cocoon (grey oval within the blue) (adapted from Meusch et al, 2014). **(B)** The purified BC3-C3bot fusion protein was separated on an SDS–PAGE gel (Coomassie blue). For the Western blot, a C3bot-specific antibody was used (Aktories et al, 1989). **(C)** Autoradiogram of in vitro ADP-ribosylation of RhoA by 3 nM of BC3-C3bot and 1.25 nM C3bot (WT). **(D)** Intoxication of HeLa cells with 10 nM of TcA + BC3-C3bot each for 6 h at 37°C. **(E)** Quantification of intoxication of HeLa cells treated as in (D), plus 100 nM of C3bot (WT). **(F)** TEER assay of CaCo-2 cells treated with 1.94 nM of TcA and 5.1 nM of BC3-C3bot for 8 h. Unpaired, two-tailed *t* test (*P* < 0.001) (±SEM). Scale bar: 100 $\mu m$. N = 3.

TcB2 and TcC3 [Meusch et al, 2014]) and the chimeric complex (TcB-TcC3$_{N-term}$-C3bot [BC3-C3bot]) was expressed in *Escherichia coli* (Fig 1A). As expected, the recombinant toxin chimera was auto-proteolytically cleaved (Fig 1B) into the 25-kD C3bot and a 246-kD fragment (TcB-TcC3$_{N-term}$). C3bot ADP-ribosylates the small GTPase RhoA, thereby inactivating it. To study correct folding and activity of the fusion toxin, we performed in vitro ADP-ribosylation assays with recombinant RhoA and radiolabeled NAD as a co-substrate in a time-dependent manner. As shown in Fig 1C, the fusion toxin (BC3-C3bot) was able to modify RhoA. However, compared with C3bot alone (Fig 1C, bottom), activity of the fusion toxin (Fig 1C, top) was much weaker, most likely, because in the chimeric fusion protein, C3bot is packed within the BC3 cocoon.

To gain insight into the question of whether the cocoon shields the toxin from its substrate and/or co-substrate, we incubated *Photorhabdus* BC3 at elevated temperatures (4°C, 40°C, and 95°C) to destroy the cocoon, cooled the samples down to 21°C again, and performed in vitro ADP-ribosylation of actin. Interestingly, pre-incubation of BC3 at 95°C led to the highest activity (Fig S1A), whereas C3hvr alone (Fig S1B) showed no temperature dependence and was stable up to 95°C. The data indicate that the cocoon shields the catalytic C3 toxin from its substrate. After artificial opening upon temperature elevation, the heat-stable C3hvr was most likely released.

The most crucial question of the engineered protein injection machinery was its effect on living cells. To analyse effective transport of C3bot by the recombinant toxin complex, HeLa cells were incubated with TcA (PTC subunit TcdA1), BC3-C3bot, TcA plus BC3-C3bot, recombinant C3bot alone, or were left untreated. As

shown in Fig 1D, only cells treated with both components of the toxin complex rounded up, indicating RhoA inactivation. This suggests functional delivery of C3bot by the protein complex into HeLa cells (Fig 1D, quantification shown in Fig 1E). To confirm the results observed with HeLa cells, which are cervical cancer cells, we also tested an intestinal epithelial cell type using colon carcinoma (CaCo-2) cells. Inactivation of RhoA increased permeability of ep-ithelial junctions reflected by a drop in trans-epithelial electrical resistance (TEER) (Gerhard et al, 1998). The cells were seeded on filter culture inserts and grown to confluence. The cells were then incubated with each single toxin component or with the full complex. After different incubation times, TEER was measured. Consistent with the analyses of HeLa cells, the resistance built by the epithelial barrier exclusively dropped in the presence of the full toxin complex (Fig 1F). The data show that *C. botulinum* C3 was injected by PTC as foreign cargo into mammalian cells.

## Preconditions for functional delivery

The enclosed tripartite toxin structure carries the active enzymatic component of the toxin sheltered from the environment. Structure analysis revealed a negatively charged inner surface of the cocoon (Gatsogiannis et al, 2016; Gatsogiannis et al, 2018), which suggests that the cargo is positively charged. To gain insight into the features of the cargo, which are required for effective packing and transport, we first compared the properties of naturally occurring substrates of the nanomachine, although not all are characterized yet. As shown in Table 1, all putative proteins injected by the PTC have a similar molecular weight (between 28 and 35 kD). Remarkably, their

**Table 1.  Properties of *Photorhabdus* proteins (in grey) injected by the PTC and various proteins selected for characterizing the complex's translocation capabilities.**

| Gene/Protein name | Activity | MW (kD) | pI | Reference |
|---|---|---|---|---|
| TccC3hvr_W14 | ADP-ribosyltransferase | 32 | 9.68 | Lang et al (2010) |
| TccC2_W14 | Unknown | 28 | 8.7 | |
| TccC4_W14 | Unknown | 30 | 6.23 | |
| TccC5_W14 | ADP-ribosyltransferase | 30 | 8.67 | Lang et al (2010) |
| TccC1_TT01 | Unknown | 35 | 11.01 | |
| TccC2_TT01 | Unknown | 28 | 9.4 | |
| TccC3_TT01 | Unknown | 32 | 9.06 | |
| TccC4_TT01 | Unknown | 32 | 9.38 | |
| TccC5_TT01 | Unknown | 30 | 9.08 | |
| TccC6_TT01 | Unknown | 32 | 8.9 | |
| TccC7_TT01 | Unknown | 28 | 8.56 | |
| YopT$^{\Delta 1-74}$ | Cysteine protease | 30 | 6.2 | Shao et al (2003) |
| YopT$^{\Delta 1-74}$ – lys3 | Cysteine protease | 30 | 8.29 | Shao et al (2003) |
| YopT$^{\Delta 1-74}$ – lys6 | Cysteine protease | 30 | 8.91 | Shao et al (2003) |
| YopT$^{\Delta 1-30}$ | Cysteine protease | 30 | 9.04 | Shao et al (2003) |
| YopT full | Cysteine protease | 36 | 8.53 | Shao et al (2003) |
| C3bot | ADP-ribosyltransferase | 25 | 9.57 | Aktories et al (1988) |
| *Metridia* luciferase 7 | Luciferase | 16.5 | 8.66 | Markova et al (2019) |

isoelectric point (with one exception) is above 8.5. Our artificial cargo, C3bot, shows similar characteristics with an even lower molecular weight (25 kD) and a pI of 9.57. The comparison of the open reading frames of possible *Photorhabdus* toxins delivered by PTC and C3bot suggests that the size and the charge of the cargo are key for effective transport. However, so far, exclusively ADP-ribosyltransferases and proteins with unknown activities were included in our studies. For further analyses, we chose the *Y. enterocolitica* toxin YopT. The cysteine protease is injected by type-III secretion into mammalian cells to cleave and inactivate Rho GTPases. We generated *Photorhabdus* BC3 fused to YopT proteins of different sizes and isoelectric points (compare Table 1). Besides full-length YopT, we cloned the smaller N-terminal truncations YopTdelta1-30 and YopTdelta1-74 which possessed full activity and stability when expressed as recombinant protein (Sorg et al, 2003).

The acidic charge of YopTdelta1-74 (pI = 6.2) was varied by the C-terminal addition of three or six lysine residues (with resulting pIs of 8.29 and 8.91, respectively).

## Functional injection of YopT proteins into HeLa cells

We first checked whether the chimeric proteins are catalytically active by analysing the release of RhoA from purified cell membranes. Therefore, membranes were incubated with or without the BC3-YopT proteins, separated into pellet and supernatant by ultracentrifugation, and analysed for the presence of RhoA in each fraction. Incubation of the membranes with functional YopT should release cleaved RhoA from the membranes. As shown in Fig 2A, all constructs showed YopT activity. Heat-inactivated enzyme was not able to cleave and release RhoA. Next, we analysed the functional

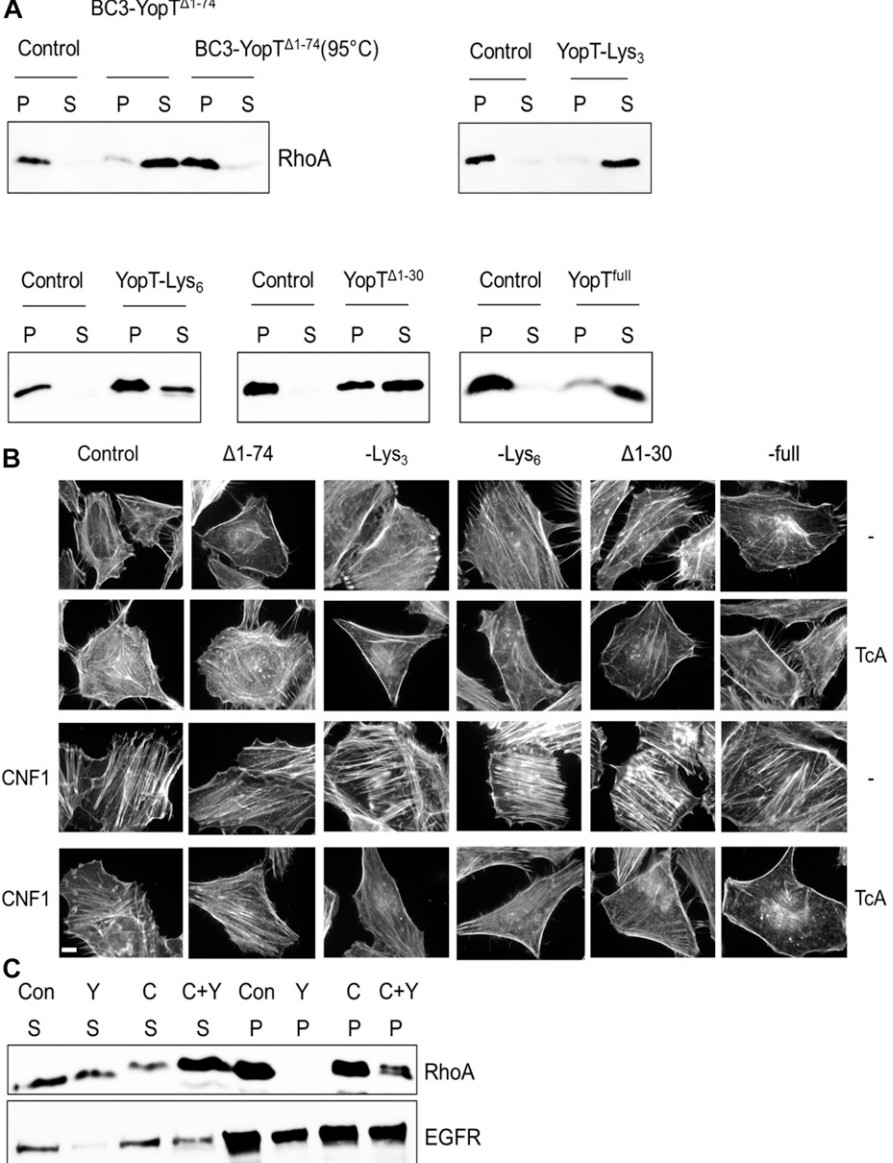

**Figure 2. Activity of BC3-YopT fusion toxins.**
**(A)** In vitro membrane release activity of RhoA from purified HeLa cell membranes by BC3-YopT chimeras. **(B)** In vivo activity of BC3-YopT. HeLa cells were intoxicated either overnight (top two panels) or for 4 h with TcA + BC3-YopTs (20 nM) (bottom two panels), then 1 h with CNF1 (4 nM) before being fixed and actin stained. **(C)** Biochemical analysis of in vivo membrane release of RhoA by BC3-YopT$^{full}$ in HeLa cells intoxicated as in B above. Scale bar: 100 $\mu$m. N = 3. P, Pellet (membrane fraction); S, post-membrane supernatant.

delivery of YopT and YopT truncations by the toxin complex into HeLa cells. Therefore, the cells were incubated with the BC3-YopT constructs in the presence or absence of TcA, respectively. After overnight incubation, the cells were fixed and stained for F-actin with rhodamine–phalloidin. As shown in Fig 2B (top two panels), artificial injection of YopT led to diminished formation of actin stress fibres, indicating inactivation of RhoA-dependent signalling pathways. Compared with the other constructs, BC3-YopTdelta1-74 had little or no activity, although it was able to cleave RhoA from purified membranes (Fig 2A). In contrast, addition of three or six lysines changing the pI of the cargo led to injection of YopTdelta1-74. Incubation of cells with TcA only (control) had no effect. As we have shown before, the YopT phenotype could not be reverted by the bacterial toxin cytotoxic-necrotizing factor 1 (CNF1) (Sorg et al, 2001). CNF1 leads to constitutive activation of Rho proteins by deamidation resulting in strong stress fibre formation. However, mis-localization of the activated Rho protein after YopT injection led to dominant destruction of stress fibres (Sorg et al, 2001). Therefore, YopT-induced destruction of stress fibres should be more visible when cells are co-treated with CNF1. As expected, injection of YopT led to diminished formation of stress fibres in the presence of CNF1 (Fig 2B, bottom two panels). Again, incubation with BC3-YopTdelta1-74 in the presence of TcA as well as incubation of cells with TcA only had no effect. In contrast to the stress fibres, polymerised cortical actin was still present after YopT delivery. The data indicate that the YopT proteins were injected into HeLa cells and that besides the size of the protein, the charge would be essential for functional delivery.

For further proof of YopT activity inside the cells, we studied cleavage and mis-localization of RhoA by separation of membranes and cytosol of cells incubated with the respective toxin complex proteins (Fig 2C). These results mirrored the previous ones, with a complete release of RhoA from membranes of YopT-treated cells and a reduction of RhoA release in cells treated with YopT plus CNF1, further confirming successful delivery of YopT into the cells.

## Time-resolved analysis of YopT action

As shown before, inactivation of Rho GTPases increased the permeability of epithelial junctions (Zihni, 2014). However, the effect of recombinant YopT on tight junctions has not been studied because of missing transport into cells. Using the PTC-mediated injection of YopT, we compared the activity of the YopT constructs by analysing their effect on Caco-2 (human colon carcinoma) cells in a time-resolved manner. Confluent CaCo-2 cells (TEER = 100%) were treated with increasing concentrations from 1 to 30 nM of each toxin chimera. Transepithelial resistance was measured continuously for up to 20 h. PTC3 was used as positive control and expectedly decreased the epithelial barrier within a few hours. As shown in Fig 3A, in the presence of TcA, BC3-YopTdelta1-74 did not change the transepithelial resistance of the monolayer, which is consistent with the HeLa cell experiments, indicating that YopTdelta1-74 was not actively injected. In contrast, together with TcA, all other BC3-YopT constructs increased the permeability of epithelial junctions with comparable activity (Fig 3B–E). Fig 3F shows a direct comparison of all YopT chimeras with a constant concentration of 30 nM. The effective concentration (EC50) of all constructs was similar

(15–20 nM), with BC3-YopTd30 being the most effective protein (compare Table 2 and Fig S2). The data indicate that YopT was injected by the *Photorhabdus* injection machinery into CaCo-2 cells. Moreover, YopT was sufficient to destroy the epithelial barrier by mis-localization of Rho GTPases. PTC-based YopT chimeras, therefore, may now be used for biochemical analysis of recombinant YopT because the toxin was injected most likely into all cells of a culture. However, the YopT data do not allow for quantification of the injected cargo molecules.

## Quantitative analysis of injected luciferase

Only few molecules of injected toxins are required for complete modification of substrate and the subsequent morphological changes of the cells. Therefore, the highly active toxins are not feasible for exact quantification of the cargo injected into each cell. Based on the composition of the nanomachine, each toxin complex injects a single enzyme into the cell. However, how many PTCs bind to each single cell and functionally release the load remains an open question. To gain insight into the number of molecules injected, we made use of a secreted luciferase produced by the marine copepod *M. longa* (M-Luc7), which was cloned 15 years ago (Markova, 2004). This enzyme encompasses all requirements needed for functional packing and injection. M-Luc7 (cloned without the signal peptide) has a small molecular weight (152 aa, 16.5 kD) and a basic character (pI = 8.66). It shares no sequence or structural homology to *Renilla* or firefly luciferases (for review see Markova et al (2019)) although, like *Renilla* luciferase, it uses commercially available cell-penetrating coelenterazine as a substrate for producing blue light.

After binding to mammalian cells, the PTC is taken up by endocytosis. Upon intoxication, a higher amount of luciferase is detected when cells have been incubated with the full toxin compared with the cocoon alone, indicating that the luciferase is taken up into the cells (Fig 4A). However, to distinguish between endocytosis and injection, cells pretreated with bafilomycin A1 (to block acidification of endosomes) were incubated at 4°C with only the loaded cocoon (BC3-Mluc7) or with the full toxin complex. Treatment with bafilomycin and the constant 4°C was meant to limit endocytosis, thereby restricting toxin effects to only molecules bound at the cell surface. Injection was either induced by a pH change to acidic pH (pH 5) or the cells were incubated at neutral pH 7.5 as indicated. The cells were washed and treated with trypsin/EDTA to degrade extracellular protein and simultaneously to detach the cells from the dish. The cells were harvested, lysed, and lysates transferred into a white plate for detecting luciferase activity. Similarly, at both pH 5 and 7.5, a higher amount of luciferase was detected when cells were incubated with the full toxin compared with the cocoon alone, indicating that the luciferase was delivered into the cells (Fig 4B). In acidic pH, however, much more luciferase was measured, suggesting favourable injection of the enzyme at this pH (Fig 4B, compare Fig 4D and E). Interestingly, comparing endocytic and injection-based delivery of YopT, the latter revealed much higher activity, especially at pH 5 (Fig 4A and B). As further proof of the distinction between endocytosis- and plasma membrane–based toxin deliveries, we conducted a similar experiment using the microtubule disrupting inhibitor, nocodazole. As shown in Fig S3,

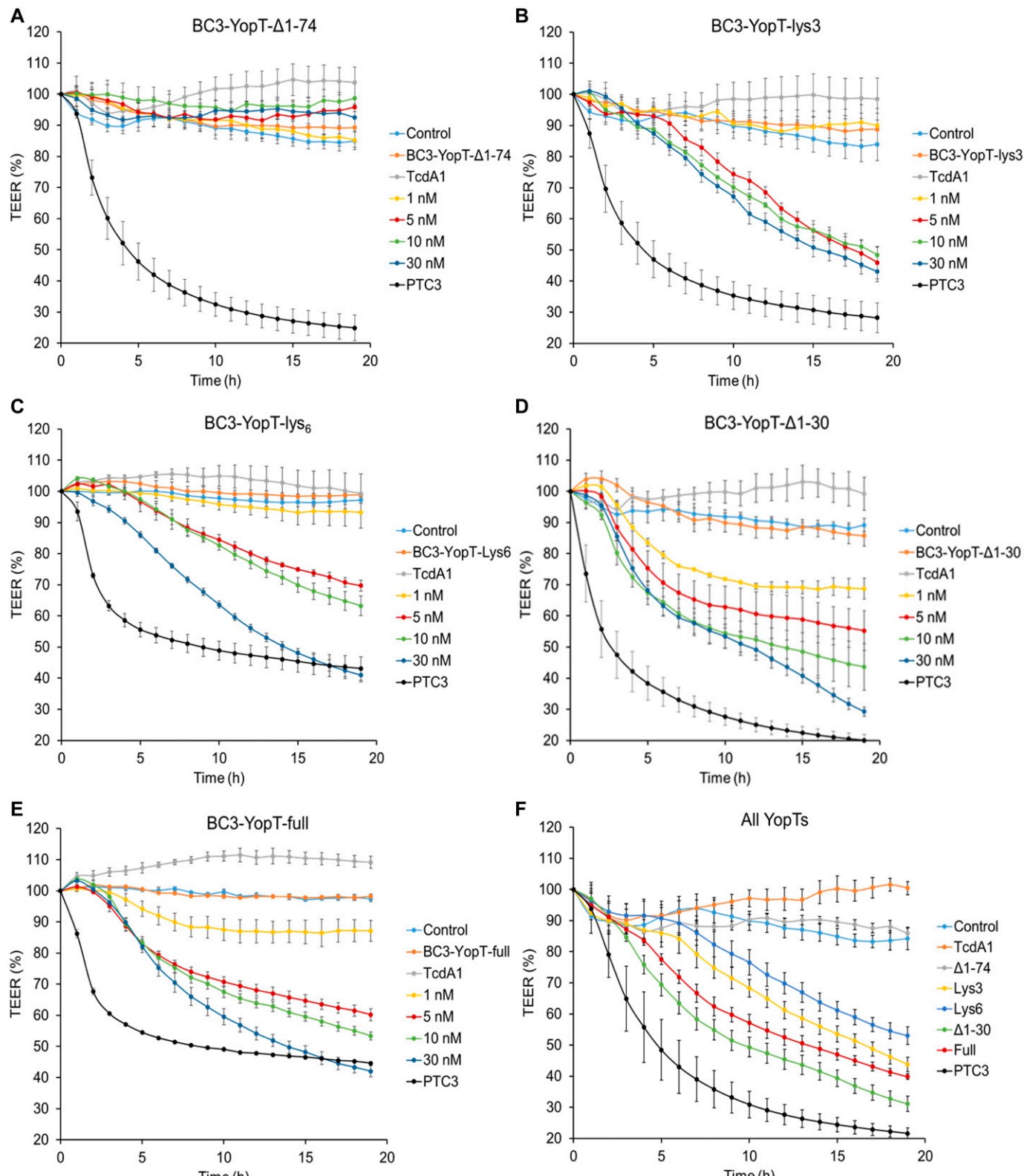

**Figure 3. In vivo activity of BC3-YopT fusion toxins.**
Confluent monolayers of CaCo-2 cells were intoxicated with increasing concentrations of TcA + BC3-YopT fusion toxins and TEER was measured. The graphs show TEER as a percentage of the starting value. **(A)** BC3-YopT$^{\Delta 1-74}$. **(B)** BC3-YopT$^{\Delta 1-74}$-lys3. **(C)** BC3-YopT$^{\Delta 1-74}$-lys6. **(D)** BC3-YopT$^{\Delta 1-30}$. **(E)** BC3-YopT$^{full}$. **(F)** A comparison of all toxins (30 nM each). Recombinant PTC3 was used as a positive control. N = 3.

**Table 2. Comparison of the EC50 values of various BC3-YopT fusion proteins after a 20-h intoxication of CaCo-2 cells in a TEER assay.**

| Toxin | EC50 |
|---|---|
| YopT$^{\Delta 1-74}$-lys3 | 17.43 |
| YopT$^{\Delta 1-74}$-lys6 | 19.7 |
| YopT$^{\Delta 1-30}$ | 8.27 |
| YopT$^{full}$ | 14.52 |

normal intoxication of HeLa cells by PTC3 (recombinant wild type), which involves endocytosis, was inhibited by nocodazole (Fig S3A) but not plasma membrane–based delivery of luciferase by PTC3-MLuc7 fusion complex (Fig S3B). In line with this, injection at the plasma membrane closely mimics the native system where the toxins are injected via the type-III secretion system.

To calculate the injected molecules per cell, a standard curve detecting luminescence produced by increasing amounts of purified BC3-MLuc7 in buffer and cell lysate was prepared (Figs 4C and S4). In addition, standard curves detecting luminescence produced by injecting MLuc7 into cells from the cell membrane at pH 7.5 (Fig 4D) and pH 5 (Fig 4E) were prepared. From these curves, we estimated about $3.4 \times 10^8$ molecules of MLuc7 produced one light unit (LU) in our system (lysate). This corresponds to 30.5 LUs in $10^6$ cells produced by $10^{10}$ molecules of MLuc7, giving $10^4$ molecules injected into each cell at pH 7.5 (961.5 LUs giving $3.1 \times 10^5$ molecules per cell at pH 5) when the cells were incubated with 1 nM of the full complex.

To further prove intracellular localization of the injected luciferase, we incubated CaCo2 cells with buffer, TcA (10 nM), BC3-MLuc7 (10 nM), or PTC3-MLuc7 (10 nM TcA + BC3-MLuc7), removed extracellular protein by treatment with trypsin/EDTA, added cell-permeable coelenterazine, and detected luminescence by live-cell microscopy. Bioluminescence images (between 450 and 700 nm) were acquired with a GaAsP detector (gallium arsenide phosphide; enabling higher quantum yields and thereby sensitivity compared with conventional photomultiplier tubes) set to maximum gain.

As shown in Fig 5, only cells treated with the full complex show a bright blue luminescence, indicating successful delivery of the luciferase.

## Discussion

Transport of proteins through cellular membranes, either for cell biological studies or for therapeutic approaches is a technical challenge for scientists. Besides manual microinjection into single cells, some methods to transport proteins through biological membranes have been developed: Cell-penetrating peptides such as Antennapedia homeodomain protein and HIV-1 Tat are able to cross cellular membranes and have been used to transport small proteins fused to it into the cells (Rizzuti et al, 2015), although the exact mechanism of membrane passage is not known. Moreover, peptidomimetics were developed to mimic the conformation of peptide sequences. However, mimicking larger protein surfaces is not yet possible (deRonde & Tew, 2015). Pore-forming bacterial toxins such as *Bacillus anthracis* protective antigen or *C. botulinum* C2 toxin have been exploited to deliver recombinant proteins across

cell membranes (Barth et al, 2002; Collier & Young, 2003). Bacteria are able to actively inject cargo proteins into mammalian cells by type-III secretion machineries (Ballard et al, 1996; Cornelis, 2000; Mechaly et al, 2012). Such bacteria have been genetically manipulated as delivery vectors for vaccination, which requires the direct contact of living bacteria with mammalian cells (Epaulard et al, 2006; Nishikawa et al, 2006; Derouazi et al, 2010). Here, we characterized a stand-alone injection nanomachine of the entomopathogenic bacterium *P. luminescens*. Physiologically, it injects different proteins into insect cells, most of them with unknown function (compare Table 1). Two of the injected toxins, however, are already characterized as ADP-ribosyltransferases modifying actin and Rho GTPases, respectively (Lang et al, 2010). We showed that the *C. botulinum* ADP-ribosyltransferase C3bot was accepted as foreign cargo. When injected by the nanomachine, the toxin induces typical cell rounding within 6 h by destruction of the actin cytoskeleton. In former studies, C3, which consists of an enzymatic domain exclusively, was delivered by pore-forming toxins such as *C. botulinum* C2 toxin. Using this kind of delivery system, uptake of C3 led to cell rounding within 2–3 h (Barth et al, 2002), suggesting a more effective transport. An advantage of the PTC over the pore-forming transporters may be the possible protection from degradation within the closed cage. However, whether protection is true for all types of cargo analysed, needs to be studied.

To study whether also other enzymes than ADP-ribosyltransferases would be packed and injected, we further fused the protease *Y. enterocolitica* YopT to BC3. YopT is physiologically injected by type-III secretion. It localizes to cellular membranes and releases mainly RhoA by cleaving off the C-terminal isoprenylated cysteine (Shao et al, 2003). Moreover, RhoA is released from GDI, leading to proteasomal degradation of the small GTPase and to disruption of stress fibres (Zumbihl et al, 1999; Aepfelbacher et al, 2003). In our study, YopT was released from endosomes. In contrast to manual microinjection or expression of YopT, experiments in which cell rounding occurred within minutes, destruction of stress fibres and opening of barriers was much slower when only few molecules of YopT were injected. This may more closely reflect the physiological situation, including substrate specificity. Although purified YopT does not show specificity for any specific Rho GTPase in vitro, spatial localization of the injected protease may be the basis for its specificity in living cells (Sorg et al, 2001). In mammalian cells, RhoA is attached to the plasma membrane, whereas Rac and Cdc42 show endosomal and Golgi localization (Michaelson et al, 2001). In *Yersinia*-infected macrophages, RhoA seems to be the preferred substrate of YopT (Aepfelbacher et al, 2003). In our experiments using the PTC for injection of YopT, cells do not round up completely. However, stress fibre formation was reduced, even in the presence of the Rho activating toxin CNF1. In former experiments, YopT was injected manually and the cells rounded up within minutes, which could not be reverted by CNF1 (Sorg et al, 2001).

In *Yersinia*, the N-terminal amino acids of YopT are involved in binding of its chaperone SycT (Buttner et al, 2005). Chaperone interaction is required for efficient translocation by the type-III secretion system (Trulzsch et al, 2004). However, the truncated recombinant proteins showed activity and high stability in in vitro assays (Sorg et al, 2003). Therefore, YopT truncations were analysed for injection by the PTC. The fragment with the highest stability and

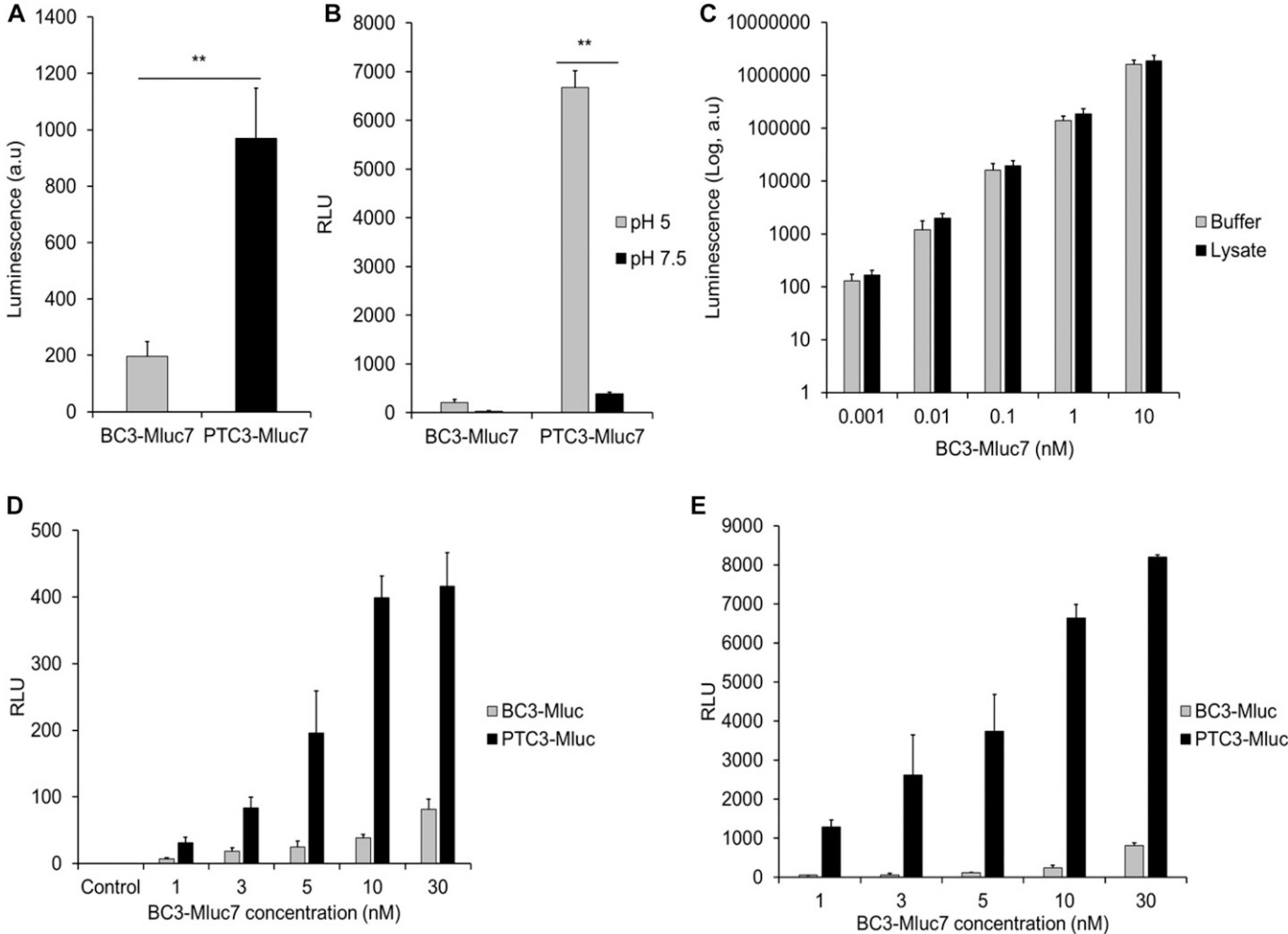

**Figure 4. Luciferase activity of BC3-MLuc7 fusion toxin.**
**(A)** In vivo activity of MLuc7 after treatment of HeLa cells with 10 nM of TcA + BC3-MLuc7 for 1 h at 37°C. **(B)** pH-dependent delivery of MLuc7 across the cell membrane of HeLa cells. **(C)** In vitro activity of BC3-MLuc7 in passive lysis buffer (Buffer) or HeLa cell lysate. **(D)** pH-dependent delivery of MLuc7 across the cell membrane at pH 7.5. **(E)** pH-dependent delivery at pH 5. Unpaired, two-tailed *t* test (*P* < 0.01) (±SEM). N = 3. RLU, relative luminescence units (luminescence per µg of total protein).

in vitro activity showed little or no effects on HeLa- and Caco-2 cells. Two reasons could be raised: unfolding of the packed cargo may be necessary for loading or the acidic pI disfavours loading into TcA. Therefore, we changed the pI by adding three or six lysines, which brought back activity, indicating that a basic pI was essential for successful injection. The interesting question of whether TccC4_W14 is delivered despite of its low pI (6.23) remains open because the activity of this toxin is still not known.

According to the assembly of the injection apparatus, each nanomachine is capable of injecting a single molecule. We, therefore, asked how many toxin complexes do bind to each single cell and are able to inject their load. To answer this crucial question, we made use of the marine luciferase *M. longa* MLuc7. The enzyme is released from the copepod to glare its predator. It does not need ATP but converts coelenterazine into light by oxidation. Because in living cells are reducing conditions, we would expect lower activity compared with buffer conditions. Our calculations, therefore, may be too low. Truly, our data can give only a rough estimate. However, we calculated about $10^4$ molecules of MLuc7 injected into each cell

at pH 7.5 and $3.1 \times 10^5$ molecules per cell at pH 5 when cells were treated with the full complex (concentration 1 nM). This leads us to suggest that 0.5% of the luciferase entered cells (0.016% at pH 7.5).

With this highly sensitive and quantifiable injected luciferase at hand, we will be able to compare the efficiency of further engineered toxin complexes to gain insight into efficient binding and pore formation. By engineering the PTC, we developed a powerful system to inject proteins, peptides, and potentially other molecules such as aptamers into mammalian cells opening new perspectives for cell biological and therapeutic approaches.

# Materials and Methods

### Materials

Cell culture medium DMEM and FCS were purchased from Biochrom, whereas McCoy's 5A medium was purchased from PAN Biotech. Cell

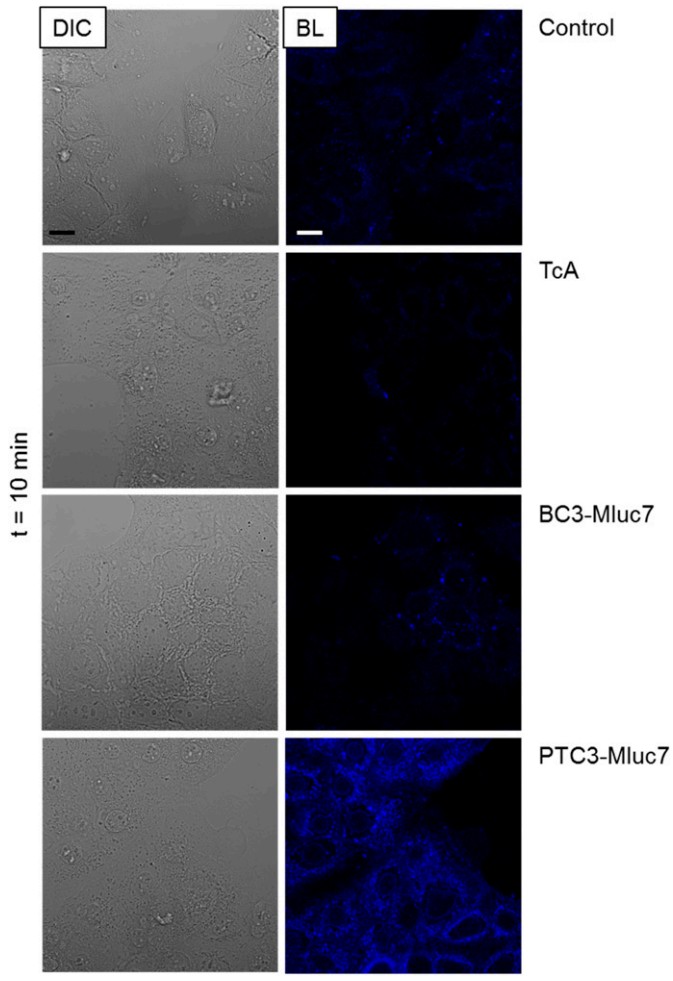

**Figure 5. Bioluminescence activity of BC3-MLuc7 fusion toxin.**
Membrane delivery of toxin (10 nM) into HeLa cells was performed for 1 h at 4°C. Then, the cells were transferred to a live cell imaging microscope, where 2 µg/ml of coelenterazine H was added before immediate visualization for 10 min. The controls include coelenterazine only, TcA + coelenterazine, and BC3-MLuc7 + coelenterazine. Figure represents one of two independent experiments. Scale bar: 15 µm. BL, bioluminescence; DIC, differential interference contrast.

culture materials were obtained from Greiner. Ni-iminodiacetic acid (IDA) resin was from Macherey-Nagel. [$^{32}$P]NAD$^+$. *Clostridium histolyticum* collagenase type-IA was purchased from Sigma (now Merck). Bafilomycin A1 was purchased from Enzo Life Sciences. Nocodazole was purchased from Sigma (now Merck). Coelenterazine H (Biotium) was purchased from Hölzel Diagnostika.

### Protein expression and purification

Proteins were expressed and purified as previously described (Lang et al, 2017). In short, *E. coli* BL21 (DE3) cells were transformed with *P. luminescens* TccC3hvr and protein expression was induced by the addition of IPTG to a final concentration of 75 µM. For *P. luminescens* TcdA1, TcdB2-TccC3, or TcdB2-TccC3 fusion proteins, *E. coli* BL21-CodonPlus cells were transformed and protein expression was induced with 25 µM IPTG. After 24 h, all *P. luminescens* protein-expressing cells were harvested and resuspended in lysis buffer

(300 mM NaCl, 20 mM Tris–HCl, pH 8.0, 1 mM DTT, 500 µM EDTA, and 10% glycerol) supplemented with DNase (5 µg/ml), lysozyme (1 mg/ml), and 1 mM PMSF. After sonication, the cell lysate was incubated with Ni-IDA resin and loaded onto empty PD-10 columns. The His6-tagged proteins were eluted with 500 mM NaCl, 20 mM Tris–HCl, pH 8.0, 0.05% Tween-20, 500 mM imidazole, and 5% glycerol. The protein-containing fractions were pooled and dialyzed against 100 mM NaCl, 50 mM Tris, pH 8.0, 0.05% Tween-20, and 5% glycerol.

For cleavage of TcdA1 by collagenase, the protein was first incubated with collagenase (3 µg of TcdA1 with 150 ng [~93 nM] of collagenase) for 1 h at 37°C in TcdA1 storage buffer without glycerol. Then, the cleaved toxin was separated from collagenase by fast protein liquid chromatography (FPLC) in TcdA1 storage buffer without glycerol as previously described (Ost et al, 2019). This involved size-exclusion chromatography with ÄKTApurifier through a Superose 6 Increase column (GE healthcare).

### Cell culture and cytotoxicity assays

HeLa and CaCo-2 cells were cultivated at 37°C and 5% CO$_2$ in DMEM containing 10% heat-inactivated FCS, 2 mM L-glutamine, 0.1 mM nonessential amino acids, 100 units/ml penicillin, and 100 µg/ml streptomycin. HT-29 cells were cultured in McCoy's 5A medium supplemented with 10% FCS and penicillin/streptomycin as mentioned above. For cytotoxicity experiments, the cells were seeded in culture dishes and incubated in medium with 0.5% FCS together with the respective toxins and/or inhibitors. After the indicated incubation periods, the cells were visualized using a Zeiss Axiovert 40CFI microscope with a Jenoptik Progress C10 CCD camera. The cytopathic effects caused by the toxins were analysed in terms of morphological changes and quantified by counting the number of intoxicated cells.

### TEER assays

For the TEER assay, the electrical cell-substrate impedance sensing system (Applied biophysics) was used. CaCo-2 cells were seeded on eight-well 8W10E+ electrical cell substrate impedance sensing arrays (Ibidi GmbH) and cultured for 2 d. Assays were performed when TEER values remained constant at ~1,000–1,600 Ω (100% confluence). First, the arrays were precooled on ice for 15 min. Then, indicated toxins were added and allowed to bind for 1 h. Finally, the arrays were transferred to 37°C where TEER was measured. The PTC3 (0.7 nM TcA + 1.75 nM BC3) was used as a positive control.

### ADP-ribosylation of actin and RhoA

For in vitro ADP-ribosylation of actin, 1.25 nM of TccC3hvr or 3 nM of BC3 was incubated with 1.9 µM βγ-actin, 150 µM NAD$^+$, radioactive [$^{32}$P]NAD$^+$ (0.5 µCi per sample) at 21°C in buffer containing 5 mM Hepes, pH 7.5, 0.1 mM CaCl$_2$, 0.1 mM ATP, and 0.5 mM NaN$_3$. After the indicated time points, the reaction was stopped by addition of SDS-containing Laemmli buffer. Then, the samples were subjected to SDS–PAGE and radiolabeled actin was detected and visualized by autoradiography. For in vitro ADP-ribosylation of RhoA, 60 nM of recombinant wild-type C3bot or BC3-C3bot fusion protein was incubated with 0.4 µM GST-RhoA, 1 µM NAD$^+$, radioactive [$^{32}$P]NAD$^+$ (0.5 µCi per sample), and 2 µg BSA at 37°C in buffer containing 25 mM

TEA, 2 mM MgCl$_2$, 1 mM DTT, and 1 mM GDP. After the indicated time points, the reaction was stopped by addition of SDS-containing Laemmli buffer, the samples subjected to SDS–PAGE, and radio-labeled RhoA detected and visualized by autoradiography.

## Membrane-release assays

For in vitro membrane release assays, HeLa cells cultured in 10-cm dishes were washed once with PBS and then detached by scrapping in 500 $\mu$l of lysis buffer (50 mM Tris–HCl, pH 7.4, 150 mM NaCl, 5 mM MgCl$_2$, 1 mM EDTA, 1 mM PMSF, and 2.5 mM DTT). The cells were then lysed by sonication and clarified by centrifugation at 1,000$g$ for 10 min at 4°C. Afterwards, membranes and cytosol were separated by ultracentrifugation at 105,400$g$ for 1 h at 4°C. The subsequent pellet, consisting of the membrane fraction, was retained, whereas the supernatant was discarded. This pellet was then resuspended in lysis buffer and either treated with 0.5 $\mu$M of each BC3-YopT fusion protein for 30 min at 37°C or left untreated and just incubated. The samples were then ultracentrifuged again to separate the pellets (membranes) from the supernatants containing released Rho GTPases (postmembrane supernatant). Finally, the samples were mixed with SDS-containing Laemmli buffer, separated by SDS–PAGE, transferred onto polyvinylidenfluoride membranes, and blotted using a RhoA-specific antibody followed by a labeled secondary antibody.

For in vivo membrane release assays, HeLa cells were first intoxicated with 20 mM of TcdA1 and BC3-YopT$^{full}$ for 4 h at 37°C, followed by 1 h with 4 nM of CNF1. Afterwards, the cells were washed, lysed, clarified, and the lysates ultracentrifuged as described above to separate membranes (pellet) from cytosol (supernatant). The pellets were then immediately resuspended in Laemmli buffer, whereas the supernatants were precipitated by addition of tri-chloroacetic acid (1:9 ratio) and incubation for 30 min at 4°C, followed by centrifugation at 16,500$g$ for 5 min. The pellets from these precipitated supernatants were then resuspended in Laemmli buffer and together with the membrane fraction, run on SDS–PAGE and blotted as above.

## Inhibition of cellular stress fibers

Four sets of HeLa cells were cultured on coverslips overnight. Then, the cells were washed with PBS and the medium changed to DMEM containing 0.5% FCS. Afterwards, two sets of cells were intoxicated for 4 h at 37°C, with 20 nM of TcdA1 together with all the indicated BC3-YopT fusion proteins. Then, one set of YopT-treated cells and an untreated set were intoxicated with 4 nM of CNF1 for 1 h. After CNF1 intoxication, the cells were washed twice with PBS, fixed with 4% paraformaldehyde for 15 min, washed again with PBS and permeabilized with 0.15% (vol/vol) Triton X-100 for 10 min. Subsequently, the cells were incubated in the dark with phalloidin–tetramethylrhodamine (TRITC) for 2 h at RT for actin staining. Finally, after washing and treatment with 70% and 100% ethanol, respectively, the cells were dried and embedded with Mowiol supplemented with DABCO (1,4-Diazabicyclo[2.2.2]octane) and DAPI for nuclear staining. Samples were cured for 24 h and the cells visualized using an Axiophot system (Zeiss).

## Bioluminescence assays

For in vitro bioluminescence assays, the indicated concentration of BC3-Mluc7 was added to 200 $\mu$l of either Passive Lysis Buffer (Promega), HeLa cell lysate, or indicated buffer. Then, this BC3-Mluc7 mix was aliquoted into a white plate and 30 $\mu$l of activity buffer (1 M NaCl, 20 mM MgSO$_4$, 0.03% gelatin, and 100 mM Tris HCl, pH 7.4) (Markova, 2004) were added. Finally, the plate was transferred to an infinite M200 microplate reader (Tecan), where 30 $\mu$l of Stop & Glo solution (Promega), diluted 1:1 with Passive Lysis Buffer, was dispensed into each well and the resultant luciferase activity detected.

For in vivo assays, HeLa cells cultured in 10-cm dishes were detached, counted, and 500,000 cells/ml taken in PBS for each treatment. The cells were incubated for 1 h at 37°C with 10 nM of either the loaded cocoon only (BC3-Mluc7) or with the full toxin complex (PTC3-Mluc7; TcA + BC3-Mluc7). Afterwards, the cells were collected by centrifugation for 5 min at 300$g$, washed, resuspended in 200 $\mu$l of passive lysis buffer, and lysed by incubating on a rotary shaker for 30 min at RT. The protein concentration of the lysate was then determined by Bradford assay, after which it was aliquoted into a white plate, activity buffer added, and luciferase activity detected as described above.

For pH-dependent assays, HeLa cells were first incubated with 100 nM of bafilomycin A1 or 20 $\mu$M of nocodazole for 30 min at 37°C. Then, they were transferred to ice, precooled to 4°C, and then incubated for 1 h with either BC3-Mluc7 only or PTC3-Mluc7 (with/without inhibitors) in HBSS buffer (1× HBSS, 20 mM Hepes, pH 7.5) at the indicated concentrations. Then, to induce injection of Mluc7 into the cells by the surface-bound toxin complex, the pH of the surrounding medium was changed either to acidic pH 5 medium (medium with 0.5% FCS and 20 mM MES, pH 5), or neutral pH 7.5 (medium with 0.5% FCS and 20 mM Hepes, pH 7.5) as indicated. These cells in pH medium were incubated for 1 min at 37°C before the medium was shifted back to pH 7.5. Afterwards, the cells were washed and treated with trypsin/EDTA for 2 min at 37°C to degrade extracellular protein and simultaneously detach them from the plate. Finally, the cells were harvested, lysed in passive lysis buffer, and transferred into a white plate for detecting luciferase activity.

For bioluminescence microscopy, CaCo-2 cells were treated as in the pH-dependent assays described above without bafilomycin A1. In contrast, these cells were not detached by the trypsin/EDTA treatment and were, thus, able to be visualized by a Zeiss LSM 800 microscope (Oberkochen). After intoxication with 10 nM of BC3-Mluc7 only or PTC3-Mluc7, 2 $\mu$g/ml of coelenterazine H diluted in PBS was added and image acquisition was started. To allow simultaneous bioluminescence and differential interference contrast imaging, the 405-nm laser was set to 0.005 mW. Bioluminescence images (between 450 and 700 nm) were acquired with a GaAsP detector set to maximum gain. Images were taken every 30 s for a total duration of 30 min. As controls, substrate only, TcdA1 only, and full toxins with YopT$^{\Delta1-74}$ and YopT$^{full}$ were used.

## Relative luminescence unit determination

To determine the RLUs, luminescence values were divided by the total protein concentration of the respective lysates. To determine

the number of luciferase molecules delivered per cell by the PTC, LUs emitted from a single cell were calculated. For this, data from in vitro bioluminescence assays were used to perform a linear regression, which generated the concentration of BC3-Mluc7 that produces one LU and the requisite number of protein molecules. Then, from the pH-dependent assays, luminescence values obtained from 1 nM of PTC3-Mluc7 were used to calculate the respective number of luciferase molecules delivered in total. A separate quantification of the number of cells per well narrowed this value down to light units emitted from a single cell, and from that, the number of luciferase molecules delivered per cell.

# Supplementary Information

# Acknowledgements

We thank Alexander Lang for fruitful discussion and Jürgen Dumbach and Antje Mueller for excellent technical support. This work was supported by the Deutsche Forschungsgemeinschaft SFB 850 (project C2 to G Schmidt). Financial support of P Ng´ang´a by the DAAD is gratefully acknowledged.

## Author Contributions

PN Ng´ang´a: data curation and formal analysis.
JK Ebner: data curation and investigation.
M Plessner: data curation and investigation.
K Aktories: supervision and writing—review and editing.
G Schmidt: conceptualization, supervision, funding acquisition, project administration, and writing—original draft, review, and editing.

## Conflict of Interest Statement

The authors declare that they have no conflict of interest.

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
