## [Reviewer comments · Life Science Alliance]

Life Science Alliance

Engineering Photorhabdus Luminescens Toxin complex (PTC) into a recombinant injection nano-machine

Peter Ng'ang'a, Julia Ebner, Matthias Plessner, Klaus Aktories, and Gudula Schmidt
DOI: <https://doi.org/10.26508/lsa.201900485>

Corresponding author(s): Gudula Schmidt, University of Freiburg

Review Timeline:

Submission Date:	2019-07-16
Editorial Decision:	2019-08-09
Revision Received:	2019-09-04
Editorial Decision:	2019-09-09
Revision Received:	2019-09-10
Accepted:	2019-09-10

Scientific Editor: Andrea Leibfried

Transaction Report:

August 9, 2019

Re: Life Science Alliance manuscript #LSA-2019-00485-T

Gudula Schmidt
University of Freiburg
Pharmakology
Albertstr. 25
Freiburg 79104
Germany

Dear Dr. Schmidt,

Thank you for submitting your manuscript entitled "Photorhabdus Luminescens Toxin complex (TCC) a recombinant injection nano-machine" to Life Science Alliance. The manuscript was assessed by expert reviewers, whose comments are appended to this letter.

As you will see, the reviewers appreciate your work and think that your conclusions are largely well supported. They note, however, that the presentation of the work needs to get revised. We would thus like to invite you to submit a revised version of your work, addressing all reviewer concerns. Flow and reader-friendliness need to get vastly improved (both reviewers) and it would be good to include another endocytosis blocking experiment (reviewer #2). Further evidence that a broad range of cargos could be delivered by the described system (reviewer #2) would be ideal, but is not mandatory for acceptance here.

Thank you for this interesting contribution to Life Science Alliance. We are looking forward to receiving your revised manuscript.

Sincerely,

B. MANUSCRIPT ORGANIZATION AND FORMATTING:

Reviewer #1 (Comments to the Authors (Required)):

This manuscript elucidates the ability of the *Photobacterium luminescens* toxin complex (PTC) to

deliver heterologous cargo polypeptides into mammalian cells. The authors report data that convincingly demonstrate the ability of the PTC to transport into cells: i) Clostridium botulinum C3 toxin, an ADP-ribosyltransferase, distantly related to two of the known PTC cargo proteins; ii) YopT, a protease produced by Yersinia enterocolitica, and N-terminal truncates thereof; and iii) M-luc7, a luciferase produced by a marine copepod. The heterologous cargo proteins are well chosen, the approaches are valid, the experiments are well controlled, and the conclusions are well supported by data. The findings demonstrate that the PTC is capable of transporting heterologous polypeptides with unrelated primary structures and enzymic properties. Results presented show a dependence of efficiency of transport of cargo on charge, cationic favoring higher efficiency; and data gathered with the luciferase permit rough quantification of the efficiency of entry. Overall, the results represent a nice advance in understanding the transport properties of a highly complex, and structurally very well characterized, system for transporting polypeptides across membranes.

My concern with this manuscript has to do with the quality of the writing and overall presentation. The manuscript needs a thorough examination with a view to correcting inaccuracies and grammatical mistakes, as well as improving stylistic factors with the aim of improving flow and reader-friendliness.

Terminology: Whether a given polypeptide should be termed a toxin vs. an effector protein vs. an enzymatic domain, etc., would require a complex discussion, but the authors should pay attention to such matters in revising the manuscript.

Title: Why is TCC use here, while PTC is used elsewhere in the manuscript?

Abstract, line 7: "...besides the size..." The authors present no data or discussion regarding the size dependence of transport.

Abstract, line 11: "...lide aptamers..." It's an interesting possibility, but this suggestion does not belong in the abstract, given the absence of data or discussion in the manuscript.

Abstract, last sentence: This sentence adds nothing; I suggest deleting it.

p 4, line 7: GDI should be defined.

p. 7, line 6: Replace "is" with "are."

p. 8, lines 6, 7: "In contrast" is used as a lead-in to two consecutive sentences.

Reviewer #2 (Comments to the Authors (Required)):

This is a nice brief report describing the engineering of Photorhabdus luminescens toxin complex (PTC) as a delivery vehicle for small, basic exogenous cargos, including small enzymes, into targeted mammalian cells. The authors show that PTC can deliver in a pH-dependent manner not only the native TcC3 ADP-ribosyltransferase cargo, but also a fusion BC3-C3bot the Yersinia T3SS effector protein YopT protease and the Metridia longa luciferase reporter protein attached to TcBC. They further determined that the charge of the cargo is important for efficient delivery by their system. Overall, the study is interesting, well-conceived and presented. There are, however, a few points that if adequately addressed would greatly strengthen the study and conclusions.

Pages 3-4, Figure 1A: Nomenclature is confusing. Description of the complex and component subunits is not clear to the audience. It is not clear where the endogenous ADPRT and protease cargo are located and where exogenous cargo is fused. What does BC3-C3bot mean in terms of protein organization? Is the C3bot cargo attached to both TcB and TcC? When you say that the reporter protein was cloned into TcBC, does that mean that the reporter was fused to the C-terminus of both TcB and TcC? When you mention BC3 cocoon, do you mean the cocoon made by the BC complex that has the TcC3 cargo attached? When you mention BC3-C3bot, do you mean that the TcC3 cargo is replaced with C3bot or that C3bot is attached onto each component of the BC3 complex (both TcC3 and TcB3)? Figure 1A suggests that there is only one TcB-TcC protein bound to the TcA pentamer, yes? Where is the autocleaving protease - in yellow or dark blue? Please make this clearer. Perhaps a labeled diagram showing all of the recombinant proteins would help.

Page 7-8: It would be helpful to refer in the text to the pI listing in the Table when discussing the pI's of the cargo. Also, state in the text what the pI is for each. For example, consider: "The acidic charge of YopT Δ 1-74 (pI = 6.2) was varied by C-terminal addition of 3 or 6 lysine residues (resulting pI = XX or YY, respectively)."

Page 8: Explain in the text the statement: "As expected, YopT-induced destruction of stress fibers was even more visible ...with CNF1..." - proved why this is an expected result.

Page 10: Regarding distinguishing between endocytosis and injection, the experiments using direct translocation at low temperature with change in pH are not very convincing. What about blocking endocytosis using colchicine or nocodazole?

Page 14: The statements "the basic pI un-favours loading into TcA" and "indicating an acidic pI was essential for successful injection" is contradictory since the addition of basic amino acid residues should be "acidic pI disfavors" and "basic pI was essential" - Lysines are basic amino acids!

Page 13: The statement "an advantage of the PTC over the pore forming transporters may be the protection from degradation within the closed cage" was based on previous studies with C3 and C3bot as cargo, but not for all of the cargo types used in this study (YopT, Mluc7). So, this is a very strong statement that has not actually been demonstrated as universal for other cargo by the authors. Since the authors are emphasizing the broad range of cargos that could be delivered by their system, it is important for them to substantiate all claims such as this for a wide variety of cargos.

Figure 5: The phrase "was first done" should be "was first performed." For the toxin treatment, specify the incubation time before treatment with coelenterazine H, as well as how long the cells were incubated after treatment before visualization. Which phase was performed for 10 min?

Throughout: It would have been very helpful if the authors had included line numbers.

Table 1: TccC4_W14 has a low pI - did it get delivered to the cytosol by PTC?

Throughout: Inconsistency in use of "BC" versus "B-C".

Throughout: "second substrate" should be "co-substrate"

Throughout: Microbe names (*Photobacterium*, *Yersinia*, *Renilla*) should be italicized.

Throughout: "induced opening of epithelial junctions" should be "increased the permeability of epithelial junctions".

Throughout: "EC50" should be "EC50 value".

Page 6: "This proves" is too strong a statement.

Page 11: Define GaAsP detectors for the reader.

Numerous syntax/grammar/typographical errors throughout. For example:

Page 3: "secretion system presuming the direct" should be "secretion system upon the direct"

Page 4: "specifically cleaving the" should be "that specifically cleaves the"

Page 5: "TcC3 towards C3bot" should be "TcC3 with C3bot", "and an about 246" should be "and a 246"

Page 6: "Analogue to the HeLa experiment, a second cell line was studied." - Awkward! Perhaps could rephrase as: "To confirm the results observed with HeLa cells, which are xxx cell type, we also tested an intestinal epithelial cell type using colon carcinoma (CaCo-2) cells." Also, "a drop of the" should be "a drop in the," "To get insight" should be "To gain insight," and "naturally appearing" should be "naturally occurring." "This suggests the cargo to be positively charged." - Awkward! Perhaps change to: "(2018), which suggests that the cargo is positively charged."

Page 7: "with and without" should be "with or without"

Page 8: "no or only little" should be "little or no", delete "In contrast,"

Page 9: delete "Therefore," "YopTwas" should be "YopT was", "E) Fig." should be "E). Fig."

Reviewer 1:

Terminology: Whether a given polypeptide should be termed a toxin vs. an effector protein vs. an enzymatic domain, etc., would require a complex discussion, but the authors should pay attention to such matters in revising the manuscript.

To avoid confusion of the readers, we now named all effectors as toxins.

Title: Why is TCC use here, while PTC is used elsewhere in the manuscript?
TCC was changed to PTC.

Abstract, line 7: "...besides the size..." The authors present no data or discussion regarding the size dependence of transport.

Our statement was changed to: We suggest that, besides the probable size limitation, the charge of the cargo also influences the efficiency of packing and transport into mammalian cells.

Abstract, line 11: "...lide aptamers..." It's an interesting possibility, but this suggestion does not belong in the abstract, given the absence of data or discussion in the manuscript.

Our suggestion to load also aptamers was deleted in the abstract section.

Abstract, last sentence: This sentence adds nothing; I suggest deleting it.
The last sentence was deleted.

p 4, line 7: GDI should be defined.

GDI is now defined as guanosine nucleotide dissociation inhibitor.

p. 7, line 6: Replace "is" with "are."

We replaced "is" with "are."

p. 8, lines 6, 7: "In contrast" is used as a lead-in to two consecutive sentences.
„In contrast“ at the beginning of the second sentence was deleted.

Reviewer 2:

Pages 3-4, Figure 1A: Nomenclature is confusing. Description of the complex and component subunits is not clear to the audience. It is not clear where the endogenous ADPRT and protease cargo are located and where exogenous cargo is fused. What does BC3-C3bot mean in terms of protein organization? Is the C3bot cargo attached to both TcB and TcC?

Each cargo is attached to the N-terminus of TcC. The catalytic hvr of TcC is replaced. There is no fusion to TcB.

When you say that the reporter protein was cloned into TcBC, does that mean that the reporter was fused to the C-terminus of both TcB and TcC? When you mention BC3 cocoon, do you mean the cocoon made by the BC complex that has the TcC3 cargo attached?

Yes, the cocoon is built by the BC complex itself.

Figure 1A suggests that there is only one TcB-TcC protein bound to the TcA pentamer, yes?

Yes, there is only one BC complex bound to the TcA pentamer.

Where is the autocleaving protease in yellow or dark blue? Please make this clearer. Perhaps a labeled diagram showing all of the recombinant proteins would help.

We are sorry about the confusion our nomenclature generates. We tried to explain it better throughout the text and changed Fig. 1 for more clarity.

Page 7-8: It would be helpful to refer in the text to the pI listing in the Table when discussing the pI's of the cargo. Also, state in the text what the pI is for each. For example, consider: "The acidic charge of YopTdelta1-74 (pI = 6.2) was varied by C-terminal addition of 3 or 6 lysine residues (resulting pI = XX or YY, respectively)."

As suggested by the reviewer, we included the pIs of the generated YopT constructs within the main text.

Page 8: Explain in the text the statement: "As expected, YopT-induced destruction of stress fibers was even more visible ...with CNF1..." proved why this is an expected result.

Our statement was explained as follows:

CNF1 leads to constitutive activation of Rho proteins by deamidation resulting in strong stress fibre formation. However, mis-localization of the activated Rho protein following YopT injection led to dominant destruction of stress fibres (Sorg et al., 2001). Therefore, YopT-induced destruction of stress fibres should be more visible when cells are co-treated with CNF1. As expected, injection of YopT led to diminished formation of stress fibres in the presence of CNF1 (Fig. 2B, bottom two panels).

Page 10: Regarding distinguishing between endocytosis and injection, the experiments using direct translocation at low temperature with change in pH are not very convincing. What about blocking endocytosis using colchicine or nocodazole?

As suggested by the reviewer, new experiments were performed using Nocodazole.

They support our statement, that injection over the plasma membrane is independent from endocytosis. The new data are included in the manuscript as new figure (S3).

Page 14: The statements "the basic pI un-favours loading into TcA" and "indicating an acidic pI was essential for successful injection" is contradictory since the addition of basic amino acid residues should be "acidic pI disfavors" and "basic pI was essential" - Lysines are basic amino acids!

Thank you for the statement. Indeed, this big mistake was missed by all authors. It is corrected in the new version of the manuscript.

Page 13: The statement "an advantage of the PTC over the pore forming transporters may be the protection from degradation within the closed cage" was based on previous studies with C3 and C3bot as cargo, but not for all of the cargo types used in this study (YopT, Mluc7). So, this is a very strong statement that has not actually been demonstrated as universal for other cargo by the authors. Since the authors are emphasizing the broad range of cargos that could be delivered by their system, it is important for them to substantiate all claims such as this for a wide variety of cargos.

Our statement was improved as follows:

An advantage of the PTC over the pore forming transporters may be the possible protection from degradation within the closed cage. However, whether protection is true for all types of cargo analysed, needs to be studied.

Figure 5: The phrase "was first done" should be "was first performed." For the toxin treatment, specify the incubation time before treatment with coelenterazine H, as well as how long the cells were incubated after treatment before visualization. Which phase was performed for 10 min?

Description of Fig.5 was corrected.

Table 1: TccC4_W14 has a low pI did it get delivered to the cytosol by PTC?

Thank you for this question. It is included in the new discussion as follows:
The interesting question of whether TccC4W14 is delivered despite of its low pI (6.23) remains open, because the activity of this toxin is still not known.

Throughout: Inconsistency in use of "BC" versus "B-C". Throughout: "second substrate" should be "co-substrate" Throughout: Microbe names (*Photorhabdus*, *Yersinia*, *Renilla*) should be italicized.

Throughout: "induced opening of epithelial junctions" should be "increased the permeability of epithelial junctions".

The mistakes mentioned were corrected in the new manuscript.

Page 6: "This proves" is too strong a statement.

„Proves“ was changed to „suggests“.

Page 11: Define GaAsP detectors for the reader.

GaAsP was defined in the new manuscript as:
Bioluminescence images (between 450 and 700 nm) were acquired with a GaAsP detector (gallium arsenide phosphide; enabling higher quantum yields and thereby sensitivity compared to conventional photomultiplier tubes) set to maximum gain.

Numerous syntax/grammar/typographical errors throughout. For example:
Page3: "secretion system presuming the direct" should be "secretion system upon the direct" Page 4: "specifically cleaving the" should be "that specifically cleaves the" Page 5: "TcC3 towards C3bot" should be "TcC3 with C3bot", "and an about 246" should be "and a 246" Page 6: "Analogue to the HeLa experiment, a second cell line was studied." - Awkward! Perhaps could rephrase as: "To confirm the results observed with HeLa cells, which are xxx cell type, we also tested an intestinal epithelial cell type using colon carcinoma (CaCo-2) cells." Also, "a drop of the" should be "a drop in the," "To get insight" should be "To gain insight," and "naturally appearing" should be "naturally occurring." "This suggests the cargo to be positively charged." - Awkward! Perhaps change to: "2018),

which suggests that the cargo is positively charged." Page 7: "with and without" should be "with or without" Page 8: "no or only little" should be "little or no", delete "In contrast," Page 9: delete "Therefore," "YopTwas" should be "YopT was", "E) Fig." should be "E). Fig."

All mistakes mentioned by the reviewer have been corrected. Thank you for the many corrections and suggestions given.

September 9, 2019

RE: Life Science Alliance Manuscript #LSA-2019-00485-TR

Prof. Gudula Schmidt
University of Freiburg
Pharmakology
Albertstr. 25
Freiburg 79104
Germany

Dear Dr. Schmidt,

Thank you for submitting your revised manuscript entitled "Photorhabdus Luminescens Toxin complex (TCC) a recombinant injection nano-machine". I appreciate the introduced changes and would thus be happy to publish your paper in Life Science Alliance. Before sending you the official acceptance letter, please:

- make sure that the author order in our submission system matches the one of your manuscript
- I would like to suggest to change the section title "statistical analysis" to "RLU determination"

A. FINAL FILES:

-- Summary blurb (enter in submission system): A short text summarizing in a single sentence the study (max. 200 characters including spaces). This text is used in conjunction with the titles of papers, hence should be informative and complementary to the title. It should describe the context and significance of the findings for a general readership; it should be written in the present tense

and refer to the work in the third person. Author names should not be mentioned.

B. MANUSCRIPT ORGANIZATION AND FORMATTING:

Sincerely,

September 10, 2019

RE: Life Science Alliance Manuscript #LSA-2019-00485-TRR

Prof. Gudula Schmidt
University of Freiburg
Pharmakology
Albertstr. 25
Freiburg 79104
Germany

Dear Dr. Schmidt,

Thank you for submitting your Research Article entitled "Engineering Photorhabdus Luminescens Toxin complex (PTC) into a recombinant injection nano-machine". It is a pleasure to let you know that your manuscript is now accepted for publication in Life Science Alliance. Congratulations on this interesting work.

DISTRIBUTION OF MATERIALS:

Again, congratulations on a very nice paper. I hope you found the review process to be constructive and are pleased with how the manuscript was handled editorially. We look forward to future exciting submissions from your lab.

Sincerely,

Andrea Leibfried, PhD
Executive Editor
Life Science Alliance
Meyershofstr. 1
69117 Heidelberg, Germany
t +49 6221 8891 502
e a.leibfried@life-science-alliance.org
www.life-science-alliance.org